# Revealing and Reducing Morphological Biases Using Implicit Neural Representations for Medical Image Registration

**Sofija Engelson**[*1] iD                                    SOFIJA.ENGELSON@DFKI.DE
[1] *German Research Center for Artificial Intelligence, Maria-Goeppert-Str. 15, 23562 Lübeck, Germany*

**Bennet Kahrs**[*2] iD                                    BENNET.KAHRS@UNI-LUEBECK.DE
[2] *Institute of Medical Informatics, University of Luebeck, Ratzeburger Allee 160, 23562 Lübeck, Germany*

**Timo Kepp**[1] iD                                            TIMO.KEPP@DFKI.DE
**Julia Andresen**[2] iD                                   J.ANDRESEN@UNI-LUEBECK.DE
**Heinz Handels**[1,2] iD                            HEINZ.HANDELS@UNI-LUEBECK.DE
**Jan Ehrhardt**[1,2] iD                                JAN.EHRHARDT@UNI-LUEBECK.DE

**Editors:** Accepted for publication at MIDL 2026

## Abstract

Deep learning has enhanced medical image analysis, yet models trained on imbalanced or non-representative populations often exhibit systematic biases, which can lead to substantial performance disparities across patient subgroups. Addressing these disparities is essential to ensure fair and reliable model deployment in clinical practice. Particularly in medical imaging, population-level biases can oftentimes be attributed to morphological rather than intensity differences, such as sex-related differences in organ volume. Given that morphological biases in neuroimaging data spuriously correlate with the disease label, we show, that bias detection based on general foundation model features (e.g., CLIP and BiomedCLIP) insufficiently captures morphological biases. Therefore, we introduce a bias detection and mitigation pipeline that performs subgroup discovery on deformation representations from a generalizable implicit neural representation (INR). This proof-of-concept study indicates improved performance when using deformation representations instead of general image features for bias detection. Furthermore, our results show that re-balancing the training dataset using the identified subgroups, complemented by INR-generated samples for augmentation, helps to mitigate the bias effect.

**Keywords:** Bias Detection, Bias Mitigation, Implicit Neural Representations, Medical Image Registration

## 1. Introduction

Deep learning has driven major progress in medical imaging, but training on imbalanced or biased datasets can cause models to exhibit systematic errors, leading to significant variations in performance among patient groups (Seyyed-Kalantari et al., 2021; Larrazabal et al., 2019). The reason behind the performance disparities across subgroups can often be traced back to shortcut learning, i.e., models rely undesirably on associated bias features

---

[*] Contributed equally

instead of causal disease attributes for classification (Brown et al., 2023). However, detecting biases in medical images is challenging, as the features that lead to spurious correlations are often complex, interdependent, and/or unknown to humans (Stanley et al., 2024).

Glocker et al. (2023) showed that subgroup differences are visible in the model's feature space by examining the principal components of features learned in the penultimate layer of a classifier and linking them to sensitive attributes like sex, race, or age. Subgroup discovery methods (SDM), such as Domino (Eyuboglu et al., 2022), address the automatic identification of underperforming subgroups based on coherent, ideally human-interpretable, image features. In doing so, they commonly utilize unsupervised clustering algorithms on a latent representation space. Within this line of work, Bissoto et al. (2025) correlate known metadata (e.g., sex, ethnicity, visual features relevant to the disease) with underperforming subgroups discovered by an SDM, and in this way aim to map subgroup disparities to human-interpretable attributes. The authors conclude that subgroups with a large performance gap do not necessarily correlate with known metadata, but that biases are often caused by subtle or non-interpretable visual features.

Especially in the medical domain, morphological features are highly predictive of certain diseases and may spuriously correlate with, for instance, gender- or age-dependent size or shape deformations (e.g., ventricular enlargement with age (Raz et al., 2005) and increased brain volume or heart morphology in men (Ruigrok et al., 2014)) or other unknown features. Even if the morphological changes are caused by attributes such as sex or age, traditional subgroup discovery fails, when metadata is not available. Stanley et al. (2025) found that shortcut learning was more heavily affected by intensity-based than by morphology-based factors, if multiple biases are present. This in turn means that, if morphological biases are present, they are more difficult to detect and, consequently, to mitigate. Previous work has mainly focused on the investigation of intensity-based biases (Sun et al., 2023; Bissoto et al., 2025), while morphological biases gained less attention. Moreover, current work neglects to offer suggestions on how to proceed with detected subgroup disparities that cannot be attributed to a known dataset characteristic.

While current SDMs rely on latent representations derived from foundation models trained on a large amount of image-text pairs (Bissoto et al., 2025) or on features learned by the investigated classifier (Plumb et al., 2023; Olesen et al., 2025), recent literature on implicit neural representations (INRs) suggests that INRs provide a rich feature encoding of the underlying data. In particular, when used as generalizable INRs – where the model is trained across a population – previews works indicate that the resulting latent space is interpretable and can be modified to learn or generate new variations of the underlying data (Großbröhmer et al., 2025; Dannecker et al., 2024). Simultaneously, recent work in deformable image registration has established INRs as a powerful representation for deformations (Wolterink et al., 2022; Kahrs et al., 2026). These findings motivate our hypothesis that deformation features can be used for downstream analysis tasks.

By investigating and addressing the above-mentioned challenges, our contributions can be summarized as follows:

- We show that concepts encoded in foundation model features (e.g., CLIP (Radford et al., 2021) and BiomedCLIP (Zhang et al., 2025)) do not sufficiently capture morphological biases.

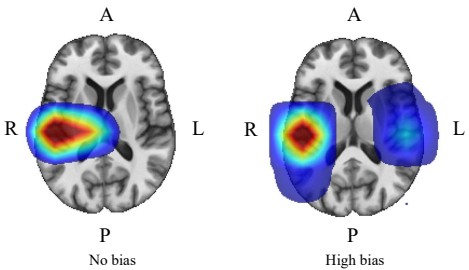

Figure 1: Grad-CAM (Selvaraju et al., 2016) heatmaps of two classifiers averaged over the true positive cases of an unbiased test dataset overlayed over the SRI24 atlas (Rohlfing et al., 2009). Classifiers trained on data with no bias and high bias are shown on the left and right, respectively. Small attribution values are discarded. Both heatmaps highlight the disease area (insula region on the right side), while the classifier trained with high bias data additionally shows activation in the bias area (putamen region on the left side).

- We propose a novel framework for the detection of morphological biases based on deformation features generated by a generalizable INR.

- We suggest a bias mitigation strategy based on discovered subgroups without knowing the attributes that lead to underperformance.

- We investigate targeted, semantically meaningful data augmentation for bias mitigation.

To enable a controlled study setup and perform a rigorous evaluation that would not be possible in a real-world scenario, we conduct a proof-of-concept study based on synthetically generated but highly realistic images, which reflect real-world deformations in neuroimaging data (Stanley et al., 2023).

## 2. Methods and Materials

Given a pre-trained classifier for binary disease classification with unknown performance and bias, the goal of this study is to reveal and reduce performance disparities due to morphological biases. We assume that the bias label spuriously and positively correlates with the disease label in the training data of the classifier. The effect of the biased vs. unbiased training data on the classifier's focus can be reviewed in the heatmaps in Fig. 1.

The general procedure of our proposed approach can be separated into two parts, that is *bias detection* and *bias mitigation*, as shown in Fig. 2. For the first step of the pipeline, we assume that the morphological characteristics of the data – i.e., natural variation, disease, and bias – can be represented by a deformation field that encodes the transformation of a given atlas image to the training images. Hence, we use generalizable INRs to encode shared morphology into a compact feature space, which, combined with classifier prediction scores, can be clustered via subgroup discovery to identify performance disparities. In the second step of the pipeline, we propose a bias mitigation strategy consisting of a subgroup

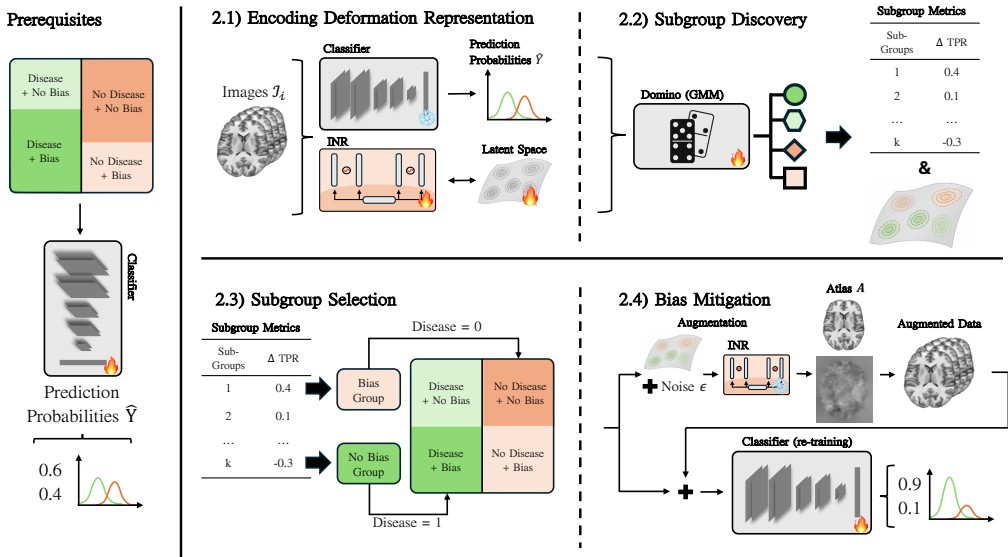

Figure 2: Schematic overview of the proposed *bias detection* and *mitigation* framework. Prerequisites are a biased dataset and an arbitrary pre-trained classifier. The bias detection is mainly shown in the upper part. The bias mitigation is visualized in the bottom part.

selection method based on theoretical assumptions, re-balancing, and data augmentation using new deformations sampled from the INR. Further, we retrain the classifier with the aim to mitigate the bias effect.

## 2.1. Encoding Deformation Representations

Given is a set of images $\mathcal{I}_i : \Omega \to \mathbb{R}$, $i = 1, \dots N$ of $N$ subjects and an atlas image $A : \Omega \to \mathbb{R}$, with $\Omega = [-1, 1]^d$ the $d$-dimensional image domain. Atlas-to-subject registration allows us to capture subject-specific morphological variations through estimated deformation fields $\varphi_i : \Omega \to \Omega$ by optimizing for $\mathcal{I}_i \approx A \circ \varphi_i$.

We follow Kahrs et al. (2026) and represent the deformations $\varphi_i$ with a generalizable INR. Unlike classical INRs, which rely on instance-wise optimization, generalizable INRs are trained across multiple instances by encoding individual information in latent representations $\boldsymbol{z}_i \in \mathcal{Z}$ that serve as additional model input (Dupont et al., 2022). Thus, the weights of the generalizable INR $\theta$ and the learnable latent representations $z_i$ are jointly trained to align the atlas image $A$ to all subject images $\mathcal{I}_i$ to obtain deformation fields $\varphi_i(\boldsymbol{x}) = f_\theta(\boldsymbol{x}, \boldsymbol{z}_i)$, $\boldsymbol{x} \in \Omega, \boldsymbol{z}_i \in Z \subset \mathcal{Z}$.

As illustrated in Fig. 3, spatial coordinates $\boldsymbol{x} \in \Omega$ and subject-specific latent representation $\boldsymbol{z}_i$ serve as the model input, and the transformed coordinates $\varphi_i(\boldsymbol{x})$ are the model output. Accordingly, the model is trained using the following loss function, which accounts for both similarity and regularization:

$$\mathcal{L} = ||\mathcal{I}_i - A \circ \varphi_i||_2 + \alpha||\varphi_i - \boldsymbol{x}||_1 + \beta\sigma(Z), \tag{1}$$

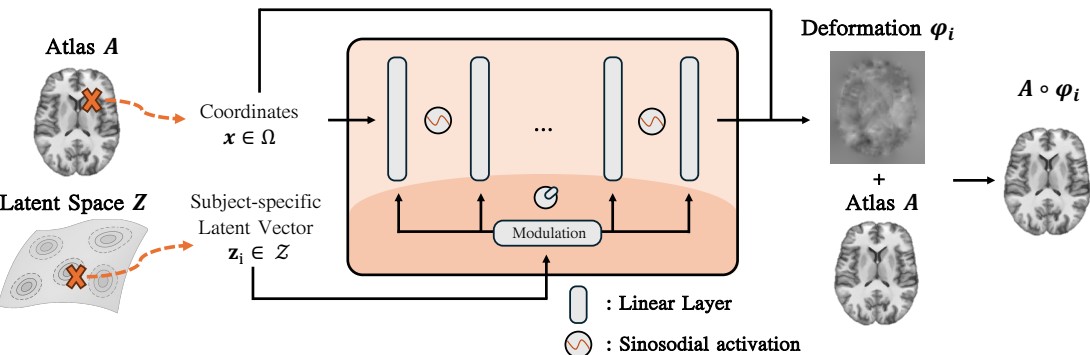

Figure 3: Structure of the generalizable INR for image registration. The INR receives spatial coordinates $x$ and a subject-specific latent vector $z_i$ as input and produces a deformation field to transform the atlas to the target patient.

where $\alpha$ and $\beta$ are weighting parameters, and $\sigma(\cdot)$ denotes the standard deviation used to regularize the latent space. The L2-norm ($||\cdot||_2$) is employed as the similarity measure due to the comparable intensity distributions across images and atlas. However, this formulation can be extended to alternative similarity metrics, such as normalized cross-correlation or mutual information, if intensity-invariant registration is required. Noteworthy – and essential to our application – is that the network parameters $\theta$ encode subject-independent information, while the latent representations $z_i$ provide a compact representation of subject-dependent morphological features and are, consistently, called deformation representations in the following. These deformation representations, learned in this first step, form the foundation for the following subgroup discovery.

## 2.2. Subgroup Discovery

The SDM Domino (Eyuboglu et al., 2022) fits a Gaussian Mixture Model (GMM) to the joint space of latent features and classifier outputs, optimizing mixture components that both form coherent clusters in the latent space and exhibit large disparities in classifier performance across those clusters. Here, we assume that the performance disparities are mainly caused by the morphological biases in the dataset. The learned deformation representations $Z = \{z_i | i = 1, \ldots, N\}$ together with the classifier softmax predictions $\hat{Y} = \{\hat{y}_i | i = 1, \ldots, N\}$ are inputs to the subgroup discovery. Domino reduces the feature representations using Principal Component Analysis (PCA) and then clusters into $K$ subgroups, $S = \{s_k | k = 1, \ldots, K\}$, where K is a hyperparameter of the method. The discovered subgroups subsequently can be utilized for more detailed performance monitoring.

## 2.3. Subgroup Selection

For bias mitigation, the goal is to create a training dataset in which the ground-truth subgroups have equal probability of occurrence. In turn, it is necessary to identify subgroups from the range of subgroups detected by Domino, that align with the underrepresented

ground-truth groups, and reweigh their occurrence probability in the bias mitigation step. Given that the bias label correlates positively with the class label, the underrepresented subgroups are *Bias & No Disease* and *No Bias & Disease*. To select the underrepresented subgroups from the subgroup clusters determined in the previous step, we calculate two metrics per subgroup: $\Delta\text{TPR}_{s_k}$ and the FPR-FNR-ratio$_{s_k}$. The subgroup performance disparity for subgroup $s_k$ is the difference between the subgroup's true positive rate and the mean true positive rate of the dataset, i.e., $\Delta\text{TPR}_{s_k} = \text{TPR}_{s_k} - \overline{\text{TPR}}$. The second metric is a relative measure based on the false positive and false negative rate of $s_k$, i.e., FPR-FNR-ratio$_{s_k} = \frac{\text{FPR}_{s_k}}{\text{FNR}_{s_k}}$. We find the bias and non-bias subgroups $S_B, S_{NB} \subset S$ based on the following assumptions:

1. The bias group has a positive $\Delta\text{TPR}$ and a high FPR-FNR-ratio.

2. The non-bias group has a negative $\Delta\text{TPR}$ and a low FPR-FNR-ratio.

The chosen metrics are based on Stanley et al. (2024) and the assumptions are derived from the following theoretical conclusions. $\Delta\text{TPR}$ examines the sensitivity differences of subgroups, hence, the high number of false negatives in the underrepresented *No Bias & Disease* group should be distinguished through a $\Delta\text{TPR}$ below the dataset mean (assumption 2). Complementarily, the FPR-FNR ratio is suitable to identify the *Bias & No Disease*, because the bias effect alone triggers false positive predictions (assumption 1). While $\Delta\text{TPR}$ has been used in previous work, we introduce the FPR-FNR ratio as a relative measure that accounts for a possible class imbalance in the subgroups, which could be obscured by absolute metrics such as $\Delta\text{FPR}$. For the metric $\Delta\text{TPR}$, we set the threshold at zero. For the FPR-FNR-ratio, we calculate the knee point over the sorted FPR-FNR-ratios of all subgroups $S$ (i.e., the point of rapid change in the FPR-FNR-ratio) and use according thresholding to identify the bias and non-bias group. To balance the underrepresented ground-truth groups (*Disease & No Bias* and *No Disease & Bias*), we remove the cases with a negative disease label from the non-bias group and the cases with a positive disease label from the bias group.

## 2.4. Bias Mitigation

Due to the defined subgroup selection restrictions in Sec. 2.3, the re-balanced dataset obtained in this step becomes small (approx. 30% of the original, biased dataset size). With the aim to increase the re-balanced dataset size, we introduce more subject variation to the re-balanced data by generating an arbitrary number of augmented samples as described in the following. We augment the training data by generating new deformations and, in this way, produce new image data by applying the deformation to the atlas image. For this purpose, we slightly vary the deformation representations and map them to dense deformation fields with our INR. Since the INR learns a continuous function from the input space to the output space, this results in similar but slightly different deformations. The challenge here is to introduce as much variation as possible while simultaneously preserving the correct disease class labels.

Therefore, we proceed as follows: Given our deformation representations $Z = \{z_i \in \mathcal{Z} | i = 1, \ldots, N\}$ and the ground-truth disease labels $Y = \{y_i | i = 1, \ldots, N\}$, we perform

PCA on $Z$ and obtain $W = [w_1, \ldots, w_q]$ with eigenvalues $\lambda_1 \leq \ldots \leq \lambda_q$ by choosing the smallest $q \in \mathbb{N}$ which explain 95% of the total variance.

We project each deformation representation in the lower dimensional space, i.e., $z_i^* = W_q^T z_i$. For each principal component, we compute the mutual information between the projected representations and the disease labels $Y$. To avoid shifts in the ground-truth label, we define $\mathcal{C}$ as a set of indices corresponding to half of the total components $q$ that minimize the mutual information.

Then, we generate new samples by adding a noise vector $\boldsymbol{\epsilon} = (\epsilon_1, \ldots, \epsilon_q)^T$ to the projected components, before back-projecting:

$$\tilde{z}_i = W_q(z_i^* + \boldsymbol{\epsilon}), \text{ with } \epsilon_j = \begin{cases} 0 & j \notin \mathcal{C} \\ \sim N(0, \sigma) & j \in \mathcal{C} \end{cases},$$

i.e., Gaussian noise is only added to components not/less associated with the disease effect. The representations $\tilde{z}_i$ are decoded by the INR, and we obtain the augmented deformation field and, consequently, with the transformed atlas, the augmented image data.

For bias mitigation, the classifier is retrained from scratch using two datasets: 1. the original, biased training dataset, and 2. a second dataset consisting of the re-balanced training dataset together with its augmented cases. Only the augmented samples that stem from the re-balanced dataset are included in dataset two, while all other augmented samples are discarded. Ideally, the second dataset comprises the exact opposite over- and underrepresentations of ground-truth groups to the first dataset, such that in total the resulting training dataset is balanced. To account for unequal sample sizes between these two datasets, we apply the Synthetic Minority Oversampling Technique (SMOTE) (Chawla et al., 2002) and assign them equal weight during training. Bias mitigation aims at increasing the recall and accuracy of the underserved, non-bias group, because missing a disease case has great consequences in health care. Moreover, a raise in sensitivity for the non-bias group, requires the classifier to look for the true causal features of the disease label.

## 2.5. Data

To enable a controlled study setup, we use SimBA (Stanley et al., 2023), a customizable framework for bias simulation in medical imaging data. This framework uses synthetic, but realistic T1-weighted MR images of the human brain, and has been used successfully in previous studies in the area of trustworthy AI (Stanley et al., 2024, 2025). In addition to natural subject variation, the framework allows simulating region-specific morphological variations that spuriously correlate with each other and, in this way, create a bias and a disease effect. We introduce morphological variation to the insula region on the right side of the brain and the putamen region on the left side of the brain for disease and bias.

Using SimBA, we generate three datasets with different bias levels, where each dataset consists of 1000 training, 500 validation, and 500 test images. The bias level, that is, the probability of bias occurrence in disease-positive samples, varies from 0.5 (unbiased), 0.7, to 0.9. As proposed by Stanley et al. (2024), the distributions of subject and disease effect magnitude are stratified across the bias levels. For evaluation, all methods are tested on the unbiased test dataset, which shows equal prevalence of all ground-truth groups (*Bias*

*& Disease, No Bias & Disease, Bias & No Disease, No Bias & No Disease*). However, the training and validation datasets inherit an unequal distribution of ground-truth subgroups.

## 2.6. Implementation Details

The generalizable INR for the atlas-to-subject registration consists of four hidden and four modulation layers with 256 nodes each. The latent size is set to 256, and the latent vectors $z_i, i = 1, ..., N$ are initialized by sampling from a normal distribution $\boldsymbol{z}_i \sim \mathcal{N}(0, 0.01)$. Both INR and latent vectors $Z$ are trained for 1000 epochs end-to-end with AdamW optimization, learning rate of 1e−5, weight decay of 1.0, and an exponentiation learning rate scheduler with a multiplication factor set to 0.999. Per training step, 16,384 two-dimensional coordinates are sampled from the image space of one subject following a normal distribution $(x, y) \sim \mathcal{N}(0, 0.4)$. In each epoch, we iterate once over all subjects in the training set. During inference, the configuration is identical; however, only the latent vectors are trained for 250 epochs, while the INR weights $\theta$ remain frozen. We use the SRI24 atlas (Rohlfing et al., 2009).

For the GMM training within Domino, we used PCA for dimension reduction to four components (explains 60% of the total variation) and set the number of subgroups $K = 7$, determined via grid search from $\{5, 7, 9, 10\}$ based on a joint analysis of classification accuracy disparities and resulting subgroup sizes. All other parameters are left at their default values. Discovered subgroups with less than ten samples are removed from subsequent processing steps. Subgroup discovery is performed for the training and validation set separately. The number of augmented samples used for bias mitigation is set to three as further increase did not improve mitigation performance.

For classifier training, we use a five-layer CNN with an exponentially increasing channel number from 16 to 256 to predict the presence of the disease label. Each convolutional block includes batch normalization and the PReLU activation function. The de-biased training and validation dataset (selected subgroups according to Sec. 2.1) are added to the original, biased validation dataset for checkpoint selection during classifier retraining.

## 3. Results

For the evaluation of our methods, we carry out experiments with training data exhibiting *strong bias* (bias level 0.9), *moderate bias* (bias level 0.7), and *no bias* (bias level 0.5). We apply the proposed subgroup selection and bias mitigation strategy to the subgroups detected by Domino on CLIP and BiomedCLIP features as comparison methods, as they are trained to cluster semantically meaningful features, and thus, are known to work well with subgroup discovery methods (Eyuboglu et al., 2022; Bissoto et al., 2025).

The third and fourth principal components of the PCA of the deformation representations of our proposed method show good separability of ground-truth groups. Fig. 4 shows the PCA-encoded deformation representations for the true disease and bias labels in the marker styles and the subgroups detected by Domino in the marker colors. The prevalence of disease and bias cases in the detected subgroups aligns well with the ground-truth groups, e.g., subgroup five (brown color) contains many disease and few bias cases and belongs mainly to the ground-truth group *Disease & No Bias* (plus sign).

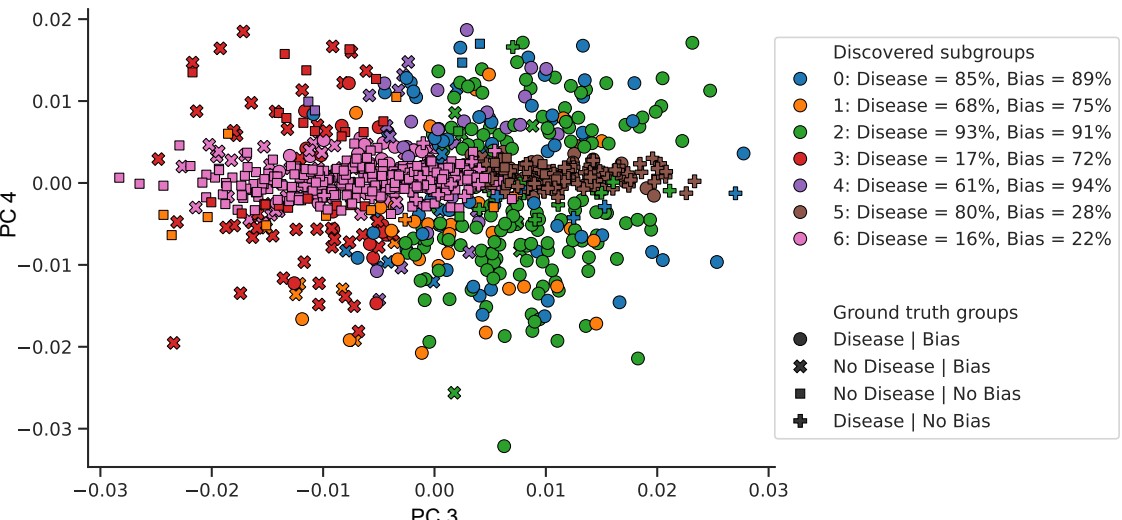

Figure 4: PCA on deformation representations (third and fourth principal component) for a dataset with bias level 0.7. Colors indicate subgroup detection with Domino, markers indicate the ground-truth disease and bias labels.

Fig. 5 shows the results for bias detection. As illustrated in the figure, the proposed method shows larger differences across subgroups than the comparison methods. Furthermore, the performance disparities detected based on the deformation representations can be associated with the increased or decreased proportion of bias or non-bias cases in the subgroup, respectively. The comparison methods show little performance deviation, and the performance disparities cannot clearly be associated with the bias label. The percentage of biased cases in the subgroups remains close to 50%, indicating that Domino was unable to separate biased and non-biased cases based on the foundation model features.

The results for bias mitigation on data trained with strong and moderate bias (Tab. 1 and Tab. 2) indicate that our proposed method significantly improves accuracy and recall for the non-bias group. While BiomedCLIP also shows significant improvements in the relevant metrics for the bias level of 0.7, this cannot be sustained when the bias level increases to 0.9. Natural for bias mitigation retraining, all methods show a decline in precision and accuracy of the bias group for bias level 0.7 (Zhang et al., 2022; Zietlow et al., 2022). The ablation study in Tab. 3 shows that the methodological components of the proposed bias mitigation approach, i.e., data augmentation as well as oversampling the new, re-balanced training dataset, show a performance gain in the relevant metrics. These improvements are further supported by a qualitative analysis of the augmented images, with two exemplary cases shown in Fig. 6. Visual examination of the generated images reveals no systematic deformation failures and predominantly realistic cases. Even rather unrealistic distortions (see Fig. 6) remain topologically and anatomically consistent, occur only sporadically, and are randomly distributed in the image space. Overall, the examples demonstrate that the applied method generates realistic subject variations while preserving anatomically intact brain MR images.

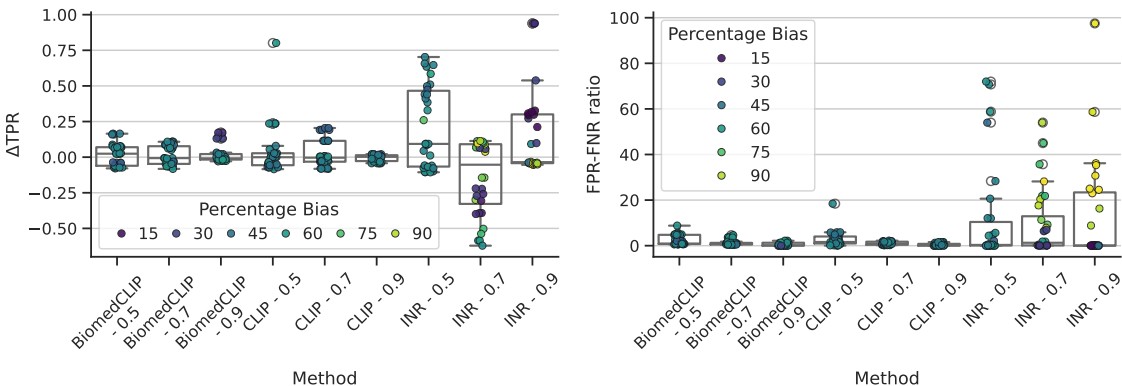

Figure 5: Results of the *bias detection* using Domino based on different models used for feature encoding and bias level for classifier training. The performance of each subgroup $s_i$ for five different seeds used to run Domino is indicated by a marker. As the number of subgroups $K = 7$, this results in maximum 35 markers. The performance differences in $\Delta$TPR and FPR-FNR-ratio are shown on the left and right side, respectively. The percentage of bias cases in the subgroup is color-coded.

| Model | Acc. B | Acc. NB | Recall B | Recall NB | Prec. B | Prec. NB |
|---|---|---|---|---|---|---|
| Unbiased clf. | $0.87_{\pm 0.01}$ | $0.87_{\pm 0.00}$ | $0.88_{\pm 0.02}$ | $0.88_{\pm 0.02}$ | $0.87_{\pm 0.02}$ | $0.87_{\pm 0.02}$ |
| Biased clf. | $0.73_{\pm 0.03}$ | $0.64_{\pm 0.04}$ | $1.00_{\pm 0.00}$ | $0.30_{\pm 0.08}$ | $0.65_{\pm 0.03}$ | $0.98_{\pm 0.01}$ |
| CLIP | $0.75_{\pm 0.02}$ | $0.67_{\pm 0.02}$ | $0.99_{\pm 0.01}$ | $0.37_{\pm 0.04}$ | $0.67_{\pm 0.02}$ | $0.97_{\pm 0.01}$ |
| BiomedCLIP | $0.76_{\pm 0.01}$ | $0.67_{\pm 0.02}$ | $0.99_{\pm 0.01}$ | $0.37_{\pm 0.05}$ | $0.68_{\pm 0.01}$ | $0.96_{\pm 0.01}$ |
| Ours | $0.75_{\pm 0.02}$ | $\mathbf{0.73}_{\pm 0.04}$ | $0.98_{\pm 0.00}$ | $\mathbf{0.49}_{\pm 0.09}$ | $0.67_{\pm 0.01}$ | $0.94_{\pm 0.01}$ |

Table 1: Results for bias mitigation with an initial bias level of 0.9 averaged over five seeds. B = bias group, NB = non-bias group. The best results of relevant metrics are marked in bold. The proposed method significantly improves accuracy NB and recall NB in comparison to the biased classifier according to a Wilcoxon rank test (p-value < 0.1), while the other methods show no significant improvement.

## 4. Discussion and Conclusion

We address the underexplored problem of morphological biases in medical imaging by presenting a framework that detects and mitigates such biases by assessing subgroup performance disparities on deformation representations learned by a generalizable INR. Using a GMM-based subgroup discovery method, we expose subgroups with biased morphological patterns, re-balance the training data using the discovered subgroups, and augment the data with synthetic samples. Our experiments demonstrate that this process effectively mitigates performance disparities, even when the related attributes are unknown. This

| Model | Acc. B | Acc. NB | Recall B | Recall NB | Prec. B | Prec. NB |
|---|---|---|---|---|---|---|
| Unbiased clf. | $0.87_{\pm0.01}$ | $0.87_{\pm0.00}$ | $0.88_{\pm0.02}$ | $0.88_{\pm0.02}$ | $0.87_{\pm0.02}$ | $0.87_{\pm0.02}$ |
| Biased clf. | $0.85_{\pm0.01}$ | $0.81_{\pm0.01}$ | $0.92_{\pm0.03}$ | $0.66_{\pm0.03}$ | $0.81_{\pm0.02}$ | $0.95_{\pm0.01}$ |
| CLIP | $0.84_{\pm0.01}$ | $0.81_{\pm0.03}$ | $0.91_{\pm0.04}$ | $0.66_{\pm0.08}$ | $0.80_{\pm0.04}$ | $0.95_{\pm0.02}$ |
| BiomedCLIP | $0.84_{\pm0.02}$ | $\mathbf{0.84}_{\pm0.02}$ | $0.95_{\pm0.02}$ | $0.75_{\pm0.06}$ | $0.78_{\pm0.03}$ | $0.93_{\pm0.03}$ |
| Ours | $0.82_{\pm0.01}$ | $0.83_{\pm0.02}$ | $0.93_{\pm0.01}$ | $\mathbf{0.77}_{\pm0.05}$ | $0.77_{\pm0.01}$ | $0.89_{\pm0.02}$ |

Table 2: Results for bias mitigation with an initial bias level of 0.7 averaged over five seeds. The proposed method and bias mitigation using BiomedCLIP features significantly improves Acc. NB and Recall NB in comparison to the biased classifier according to a Wilcoxon rank test (p-value $< 0.1$), while bias mitigation on CLIP features shows no significant improvement.

| Augm. | SMOTE | Acc. B | Acc. NB | Recall B | Recall NB | Prec. B | Prec. NB |
|---|---|---|---|---|---|---|---|
| | | $0.83_{\pm0.01}$ | $0.80_{\pm0.01}$ | $0.95_{\pm0.02}$ | $0.66_{\pm0.05}$ | $0.77_{\pm0.01}$ | $0.92_{\pm0.03}$ |
| ✓ | | $0.84_{\pm0.01}$ | $\mathbf{0.83}_{\pm0.01}$ | $0.89_{\pm0.02}$ | $0.73_{\pm0.03}$ | $0.82_{\pm0.01}$ | $0.91_{\pm0.02}$ |
| | ✓ | $0.84_{\pm0.03}$ | $0.82_{\pm0.01}$ | $0.92_{\pm0.02}$ | $0.72_{\pm0.05}$ | $0.79_{\pm0.03}$ | $0.91_{\pm0.05}$ |
| ✓ | ✓ | $0.82_{\pm0.01}$ | $\mathbf{0.83}_{\pm0.02}$ | $0.93_{\pm0.01}$ | $\mathbf{0.77}_{\pm0.05}$ | $0.77_{\pm0.01}$ | $0.89_{\pm0.02}$ |

Table 3: Ablation study for bias mitigation, exemplarily at bias level 0.7. Augm. = augmented data.

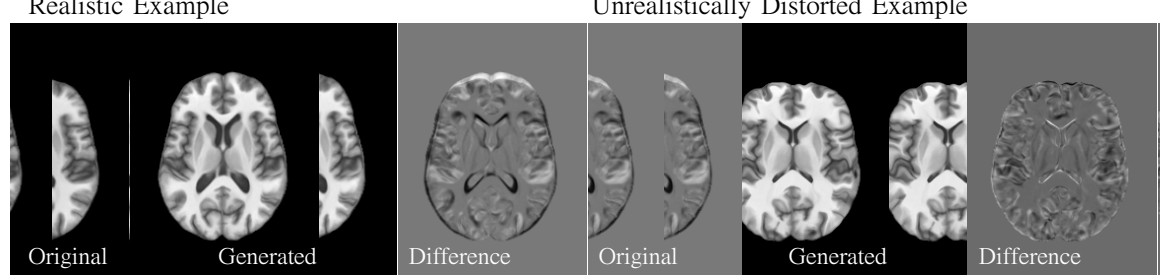

Figure 6: Qualitative analysis of augmented images. The left panel shows a visually realistic example, while the right panel shows an unrealistically distorted augmented case. The difference image indicates regular deformations, introducing subject variation rather than bias or disease-related deformation.

work acts as a proof-of-concept study on synthetically generated data to enable a controlled study setup and evaluation, which is not possible in real-world scenarios. However, the proposed methodology will be extended to real applications in subsequent research.

We demonstrate that deformation representations obtained from a generalizable INR are better suited for morphological bias detection than foundation model features encoded

by CLIP or BiomedCLIP. At a bias level of 0.9, our method achieves the best accuracy and recall for the non-bias group, while the performance gap to the unbiased classifier leaves room for improvement. With a high bias level, bias detection is performed on a small amount of underrepresented subgroups, which in turn are available for re-balancing. The proposed latent augmentation strategy introduces meaningful subject variance to the training data and artificially enlarges the re-balanced training data. While this approach may risk introducing label noise to the dataset, the ablation study shows that incorporating augmented data for retraining the classifier improves the performance.

For subgroup selection, we proposed two metrics: $\Delta$TPR and the FPR-FNR-ratio. Although these metrics are derived from theoretical assumptions, practical observations in Fig. 5 show contrastive effects for $\Delta$TPR for some subgroups. $\Delta$TPR is suitable to identify the under-represented *No Bias & Disease* group, because the large amount of false negatives leads to a negative $\Delta$TPR. However, especially with increasing bias level, the sample size of this group becomes small and more difficult to separate from the overrepresented *Bias & Disease* (positive $\Delta$TPR) group in subgroup discovery. Hence, we conclude that the contradictory empirical results are attributable to failures in subgroup discovery rather than to shortcomings of the metric itself. This indicates that further research is necessary to improve SDMs like DOMINO. High-bias subgroups suffer from many false positives, which were well captured by the FPR-FNR-ratio for all bias levels. We understand our work as a stepping stone towards further investigation and refinement of assumptions and metrics for robust bias mitigation strategies.

Fig. 4 illustrates the expressiveness of the INR's latent space, where principal components can disentangle disease and bias-related variations. With further research on component selection, we believe this method enables intuitive visualization of morphological bias patterns in the image domain and improves interpretability of unknown bias attributes. As INRs continue to evolve, their capacity for disentangling and interpreting latent representations is likely to enforce future bias detection capabilities. By leveraging subgroup discovery based on deformation representations from an INR and semantically meaningful augmentation for bias mitigation, this study makes an important step towards revealing and reducing morphological biases – even when their underlying attributes remain unknown.

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
