# OpenReview forum: "Revealing and Reducing Morphological Biases Using Implicit Neural Representations for Medical Image Registration"
_MIDL.io/2026/Conference — MIDL 2026 Poster_

### Official Review · Reviewer_69Re · 2026-01-06

**Confidence:** 3
**Preliminary Rating:** 4
**Final Rating:** 4

**Summary:**

This proof-of-concept paper tackles the underexplored challenge of morphological bias in medical image classification, which can lead to performance disparities across patient subgroups. The authors argue that standard foundation-model features (CLIP/BiomedCLIP-based) are better at picking up intensity-based biases but struggle with subtle morphological shortcuts. To address this, they propose a pipeline that uses generalizable Implicit Neural Representations (INRs) to encode deformation fields into a compact latent space, and then uses these deformation representations to identify under-performing subgroups and mitigate bias through targeted data augmentation and data re-balancing. Experiments on synthetic brain MR data show the proposed method yields a clear improvement for under-served non-bias groups compared to foundation-model-based baselines.

**Strengths:**

1. This work tackles an important but underexplored issue (i.e., morphological biases) in medical image classification. The idea of using generalized INR to represent the deformation and also as a generation tool to synthesize new images for bias mitigation is quite novel.
2. The study doesn't just stop at bias detection. It provides a concrete path for mitigation via data augmentation and subgroup re-balancing.

**Weaknesses:**

1. The major limitation is that the entire study is performed on synthetic data. While this enables controlled experiments, it leaves a big question mark on how the pipeline will handle real-world images with much more diverse anatomy and the added complexity of other bias sources. Also, deformable registration itself is challenging, and the study only involves relatively simple deformations (e.g., variations in the right insula and left putamen).
2. The pipeline depends on atlas-to-subject registration via a generalizable INR, which may be expensive and sensitive to registration quality. The paper does not report runtime, and it is hard to judge the quality of the registration process from the current presentation.
3. The bias mitigation relies on generating augmented training samples by perturbing deformation representations and decoding them through the INR and the paper mentions potential label-noise risk. However, there is not enough information to judge the quality of the generated deformations and augmented samples.

**Detailed Comments:**

It would improve clarity and reproducibility to describe the training process of the generalized INR for registration more explicitly, including the loss function(s) used and how training across subjects is carried out (e.g., how $\theta$ and $z_i$  are updated, sampling strategy, etc.).

**Justification Of Final Rating:**

The authors have provided adequate responses to my questions, and the registration and augmentation results look convincing. I agree with the other reviewers that, in its current form, it is still difficult to judge the method’s real-world applicability. Overall, I think this is a unique idea that is worth further discussion at the conference.

**Justification Of The Preliminary Rating:**

Given the challenging nature of the problem, this work is a proof-of-concept and purely based on synthetic data. But it presents a complete pipeline for morphological bias detection and mitigation. The approach is novel and worth further exploration. My main concern is whether the registration and augmentation steps remain applicable and reliable in more realistic settings.

**Questions To Address In The Rebuttal:**

1. Could the authors provide more details on the registration framework, including:
a. What loss function is used?
b. How is the generalized INR trained (especially across subjects)?
c. Are there metrics or quality control (QC) steps used to validate registration quality (and if so, can you report them)?
2. Is there any quality control or sanity check on the generated deformations and augmented samples used for mitigation? For example, a small qualitative panel showing representative augmented images and deformation fields, or examples of good and poor generated cases, or some QC statistics would be beneficial to demonstrate that the generated samples are realistic.

---

> ### Author Response · Authors · 2026-01-23
>
> We thank the reviewer 69Re for her/his questions regarding further
> details on the registration framework. A point-by-point answer can be
> found below. Details on the registration framework
>
> In response to the reviewer's comments, we have added the formulation of the loss function and clarified the training process in method and implementation details section as follows: "Accordingly, the model is trained using following loss function, which accounts for both similarity and regularization:
> $$\mathcal{L} =||\mathcal{I}_i - A \circ \varphi_i||_2 +  \alpha ||\varphi_i - \mathbf{x}||_1 + \beta \sigma(Z)$$
> where $\alpha$ and $\beta$ are weighting parameters, and $\sigma(\cdot)$ denotes the standard deviation used to regularize the latent space. The L2-norm ($||\cdot||_2$) is employed as the similarity measure due to the comparable intensity distributions across images and atlas. However, this formulation can be readily extended to alternative similarity metrics, such as normalized cross-correlation or mutual information, if intensity-invariant registration is required." (Sec. 2.1) and "Each epoch iterates once over all subjects in the training set." (Sec. 2.6) This work primarily focuses on the interpretability of the INR's latent space, while the absolute registration performance plays a secondary role. However, for bias analysis, it is important that registration errors are equally distributed across subgroups (see: answer 2 for Reviewer yv9p). Therefore, we report selected metrics (MSE and percentage of negative Jacobian determinant) for the ground-truth subgroups in the table below and confirm that the evaluated metrics are comparable across all subgroups.
>
> | Group          |           MSE                 |  Neg. Jac. Determinant  |
> | ------------ | ------------------------- | -----------------------------  |
> | No Disease | $0.001718 \pm 0.000674$  |    $0.000323 \pm 0.001232$ |
> | Disease  | $0.001834 \pm 0.000705$   |   $0.000295 \pm 0.001073$ |
> | No Bias  | $0.001664 \pm 0.000684$   |  $0.000305 \pm 0.001169$ |
> |  Bias  | $0.001889 \pm 0.000683$     | $0.000312 \pm 0.001141$ |
>
> **Table 1:** Registration performance stratified by Disease and Bias class.
>
>
> ## Quality control on the generated deformations
>
> As suggested, we have added an additional panel providing a qualitative analysis of the
> augmented images and explicitly referenced it in the result section:
> "These improvements are further supported by a qualitative analysis of
> the augmented images, with two exemplarily cases shown in Fig. 6. Visual
> examination of the generated images reveals no systematic deformation
> failures and predominantly realistic cases. Even rather unrealistic
> distortions (see Fig. 6) remain topologically and anatomically
> consistent, occur only sporadically, and are randomly distributed in the
> image space. Overall, the examples demonstrate that the applied method
> generates realistic subject variations while preserving anatomically
> intact brain MR images. " (Sec. 3) All augmented images were visually
> inspected.

---

### Official Review · Reviewer_yv9p · 2026-01-08

**Confidence:** 4
**Preliminary Rating:** 4
**Final Rating:** 4

**Summary:**

The authors address morphological bias in medical imaging in the form of systematic shape differences (e.g., organ size) that act as confounders and are often missed by foundation models. They propose using Implicit Neural Representations (INRs) to model the deformation field required to register a shared atlas to a subject. The resulting latent codes serve as features to detect biased subgroups via the subgroup discovery method Domino. Additionally, the authors introduce a mitigation augmentation strategy that generates synthetic, balanced training samples by perturbing the biased latent codes to deform the atlas. Validated on the synthetic SimBA dataset, the framework outperforms intensity-based baselines (CLIP, BiomedCLIP) in identifying and reducing morphological bias.

**Strengths:**

* The core idea of using deformation fields as the primary feature for bias detection is innovative and well-motivated for medical imaging. While most bias detection methods rely on intensity-based or semantic features (like those from CLIP), this work explicitly targets the structural variations that often drive medical bias (e.g., sex-related organ size differences).
* The paper proposes a complete pipeline that includes latent space augmentation. Leveraging the continuous nature of the INR latent space to generate new, plausible deformations for underrepresented groups is an interesting application of the technology.
* The use of the SimBA framework allows for a rigorous, controlled evaluation of different bias levels. This setup effectively isolates morphological bias, proving that the method works as intended in a "lab setting" where ground truth is known.

**Weaknesses:**

* A strong limitation of the work is the exclusive reliance on synthetic data (SimBA) for the evaluation, where bias is explicitly defined as morphological deformation. It is expected that a deformation-based detector would outperform intensity-based detectors (CLIP) on a dataset constructed via deformation rules. The paper lacks validation on real-world data or heterogeneous setups where biases might be a complex mix of intensity, texture, and shape.

* The method relies entirely on registration to a specific atlas (SRI24). This introduces a new dependency: if the atlas is not representative of the population (e.g., a young adult atlas used for an elderly cohort), the "bias" detected might simply be the *distance from the atlas* rather than the target morphological bias. This limitation is not adequately discussed.

* Questionable Augmentation Assumptions. The augmentation strategy involves applying new deformations to the same single atlas image. While this creates shape variety, it creates a dataset where every augmented image has identical intensity distributions/texture (namely of the atlas). This could introduce a severe intensity bias or artifact where the classifier learns to recognize augmented images solely by their texture, potentially harming generalization to real data.

* The authors claim that because the INR learns a continuous function, perturbing the latent space results in "similar but slightly different deformations". This implies Lipschitz continuity, which is not automatically guaranteed for neural networks without specific spectral normalization or constraints. A small perturbation in z could theoretically yield a large, unrealistic warp if the mapping is ill-conditioned.

**Detailed Comments:**

* The central claim is that deformation representations are superior to foundation model features. However, this is shown only on a dataset where the bias is exclusively morphological. How would this method perform if the bias were intensity-based (e.g., scanner artifacts)? Integrating this into a general pipeline would likely require disentangling morphological from intensity biases, which is not addressed.

* In the subgroup selection (Section 2.3), the $\Delta TPR$ metric behaves inconsistently in the results. Figure 5 shows that the assumption behind ATPR holds for bias level 0.9 but fails for 0.7. Given this instability, reliance on the FPR-FNR-ratio alone might be more robust. The work could benefit from analyzing if such a simplified metric improves stability.

* Notation: In Section 2.3, the notation for subgroup selection is confusing and needs more thorough explanation. The bar notation (e.g., $\overline{\text{TPR}}$) is standard for logical "NOT", but explicit definitions for terms like "$\Delta TPR$" and "FPR-FNR-ratio" in the text would improve readability for a broad audience.

**Justification Of Final Rating:**

I would like to thank the authors for their thoughtful rebuttal. Some of my concerns have been addressed, while concerns regarding the continuity claims of the INR (Question 4: Continuity Assumption) remain. I'm still convinced of the relevance of the work and its potentially positive impact for the MIDL conference, leaving my score unchanged.

**Justification Of The Preliminary Rating:**

The paper presents a clever and well-engineered solution to a specific problem: morphological bias. The use of generalizable INRs for this purpose is novel, and the integrated mitigation pipeline is theoretically sound. However, the evaluation is circular: the method looks for deformation biases and is tested on a dataset explicitly generated to have deformation biases. The lack of real-world validation or a "mixed bias" scenario makes it hard to judge the method's practical utility. Furthermore, the reliance on a single atlas for augmentation raises significant concerns about introducing new intensity artifacts, which are not addressed. The paper is interesting but currently feels over-fitted to its synthetic problem statement.

**Questions To Address In The Rebuttal:**

1.	Atlas Bias: Your method heavily relies on the SRI24 atlas. How do you ensure that the "biases" you detect are not simply measuring the registration difficulty or distance of certain subpopulations from this specific atlas? How would the method perform if a different atlas were used?
2.	Augmentation Artifacts: In the bias mitigation step, you generate new samples by deforming the same atlas image with new fields extrapolated from latent space. Does this not introduce a strong intensity/texture bias, where all augmented samples look like the atlas? Have you investigated if the classifier is simply overfitting to the atlas's intensity profile?
3.	Real-World Applicability: You acknowledge that the synthetic setup is a "proof-of-concept". However, since the synthetic bias is purely morphological, isn't the superiority of a morphological detector trivial? Can you provide any preliminary insight or discussion on how this performs on datasets where biases are "in the wild" and not strictly defined by deformation?
4.	Continuity Assumption: You state that the INR's continuous function guarantees similar deformations from perturbed latent codes1. Did you enforce any Lipschitz constraints or spectral normalization to ensure this smoothness? If not, did you observe any broken or unrealistic anatomies during augmentation?

---

> ### Author Response · Authors · 2026-01-23
>
> We thank reviewer yv9p for her/his comments. We appreciate the critical examination of the assumptions underlying our work and address the reviewer’s concerns through further explanations and additional experiments in the following:
>
> ## Atlas Bias
> We assume that registration errors are equally distributed across all subpopulations (see our empirical analysis on registration errors for the ground-truth groups answering reviewer 69Re). Given that no subpopulation carries larger registration errors, the choice of atlas should not substantially affect the bias detection performance. We are interested in measuring relative morphological differences between subgroups rather than absolute distances to the atlas. Differences in relative distances from the atlas for certain subgroups, e.g., elderly people, are defined as bias that we want to detect.
>
> ## Augmentation Artifacts
> The proposed augmentation technique, that uses the deformed atlas image as augmented data, is suitable for the SimBA framework only. More precisely, since in SimBA subject images are generated by applying a deformation field to the atlas ($\mathcal{I}_i \approx A \circ \varphi_i$), augmentation can also be defined as: $ \mathcal{I}_i^\*  =  A \circ \varphi_i^\*  $, where the superscript $\* $ indicates the perturbation on the deformation field for augmentation. In other use cases, it would be necessary to transform the subject images to the atlas space before augmenting them, i.e., $\mathcal{I}^\* _i = \mathcal{I}_i \circ  \varphi^\* _i \circ \varphi^{-1}_i$. Through transformation in the atlas space, individual textures and intensity variations are preserved even for the augmented images.
>
> ## Real-World Applicability
> Based on the reviewer’s comment, we applied random bias field augmentation to the subject images to simulate intensity-based biases. With these images, we retrained the classifier and the INR to encode the deformation representations and ran subgroup discovery. The discovered subgroups still largely overlap with the ground-truth groups, which shows that intensity-based biases do not interfere with morphology-based bias discovery. This is reasonable, as the registration should not be affected by changes in intensities, if a corresponding similarity distance measure is selected.
> For the joint detection of intensity-based and morphology-based biases, the INR network architecture needs to be modified to be able to process intensity values, or foundation model features need to be fused with the morphological deformation representations.

---

### Official Review · Reviewer_MB6B · 2026-01-08

**Confidence:** 4
**Preliminary Rating:** 3
**Final Rating:** 4

**Summary:**

This paper proposes to use generalizable implicit neural representations (INR) for discovering under-performing subgroups due to spuriously correlated morphological biases in brain MR imaging. The authors compare using latent INR representations to CLIP and BioMedCLIP as other latent feature extraction methods, and propose a bias mitigation technique based on data augmentation using the INR to alleviate performance disparities/shortcut learning.

**Strengths:**

This paper presents a novel approach to subgroup discovery, and I appreciate that detection and mitigation is combined within the framework as it helps to “complete the story”. The method of adding noise to non-disease related PCs and then reconstructing as a form of data augmentation is unique and potentially valuable.

**Weaknesses:**

Currently, the implementation of the INR is not entirely clear. Specifically, z is described to be extracted from the INR (e.g. shown in Fig 2.1), but is also required as input to the INR (e.g. Fig 2.3, Fig. 3). How are these z representations practically both extracted and fed as input to the INR? This requires clarification.

In addition, several aspects of the results require further explanation:
- Why does the 0.5 (unbiased) setting show so much performance variation in identified clusters (Fig 5)? Shouldn’t this essentially act as the control setting, where clusters have no meaningful differences in performance – at least not to the same extent as the bias scenarios? Are these performance differences related to global subject variation, and if so, was subject variation/disease effects identical between the different bias scenarios (and stratified, as in 10.1093/jamia/ocae165)?
- How exactly the dataset was augmented as a bias mitigation strategy is not fully clear – what determines the number of augmented samples of each class that get generated? How do the discovered subgroups inform the additional data generation? The purpose and implementation of SMOTE is also not clear – what is the minority being oversampled here?
- It is also not adequately explained why most discovered subgroups have a negative deltaTPR in the 0.7–INR case but not in the others. Furthermore, why would subgroups with 45-75% higher percentage bias have deltaTPR lower than subgroups with 15-30% bias?

**Detailed Comments:**

- Introduction: I don’t think it’s necessarily true that performance disparities are *always* a result of shortcut learning - therefore would recommend changing the second sentence to “the reason behind performance disparities across subgroups can *often* be traced back to shortcut learning”

**Justification Of Final Rating:**

The revisions to the paper improve clarity significantly. This work tackles several unique and novel areas (CLIP’s ability to identify neuroimaging morphological biases, INRs as method for subgroup discovery and bias mitigation), making it a valuable contribution.

**Justification Of The Preliminary Rating:**

The proposed method is unique and has potential for identifying and addressing specifically morphological biases in brain imaging. However, various aspects of the results are currently unclear and not entirely convincing.

**Questions To Address In The Rebuttal:**

In addition to those brought up under “Weaknesses”:
- What is the purpose of having different forms of TPR and FPR metrics (i.e. difference and ratio)?
- What was the metric used in the grid search to determine the number of subgroups?
- Would clustering separately for positive and negative classes address the problem of positive and negative deltaTPR contradictions?

---

> ### Author Response · Authors · 2026-01-23
>
> We thank reviewer MB6B for her/his feedback, and we acknowledge the potential for clarification. Please find a point-by-point answer below.
> ## Weaknesses
> 1. The latent representations $Z$ are optimized jointly during the registration process and serve as input to the INR. Thus, subject-specific information is enforced to be encoded in the latent space, while the INR captures the subject-independent information. After completion of the registration training, the learned latent representations are extracted and subsequently used for the subgroup discovery method. To clarify this workflow, we adjusted Fig. 2 accordingly and added further details to the methods section (Sec. 2.1).
> 2. We confirm that the unbiased setting (0.5 bias level) should act as a control setting, and we agree with the reviewer's assumption that the performance differences are related to subject variation. We added a clarification regarding the stratification of the subject and disease effect magnitude (Sec. 2.5). Even though the performance differences between bias level 0.7 and 0.5 are similar, the identified clusters all have a bias percentage around 50%, i.e., an equal amount of bias and non-bias cases. If subsequent subgroup selection and bias mitigation are carried out, the final classification performance of the mitigated classifier is comparable to the original classifier (see: table 1 in the next comment).
> 3. Addressing the reviewer's questions, we rewrote the corresponding paragraphs in the manuscript (Sec. 2.4 and 2.6).
> 4. Theoretically, Delta TPR should help us to identify the underrepresented *No Bias & Disease* group, because this group should contain many FN, lead to a small TPR and a negative Delta TPR. However, from an empirical perspective, we find that at a bias level of 0.7, DOMINO identifies this group inconsistently across runs, depending on the random seed. With increasing bias level, this group becomes smaller in sample size and more difficult to separate in subgroup discovery. When this group does not get recognized in subgroup discovery, we observe contrastive effects to the theoretical assumptions, namely that low bias cases have a positive Delta TPR (i.e., *No Bias & Disease* and *Bias & Disease* cannot be separated and have a positive Delta TPR in total). Nevertheless, the under-represented *Bias & No Disease* group can be distinguished well using the FPR-FNR-ratio, which is sufficient to improve the recall in classification. A less detailed statement on this and the following conclusion has been modified in/added to the manuscript: "Hence, we conclude that the contradictory empirical results are attributable to failures in subgroup discovery rather than to shortcomings of the metric itself." (Sec. 4).
> ## Questions To Address in the Rebuttal
> 1. Stanley et al. (2024) propose to use difference-based metrics such as Delta TPR and Delta FPR for measuring subgroup performance disparities. From our point of view, it is important to choose metrics that represent all fields of the confusion matrix, i.e., the TPR includes TP and FN, while FPR includes FP and TN -- this is true for the metrics proposed by Stanley et al. as well as in our paper. Furthermore, in contrast to the whole datasets, the class distribution in subgroups can be heavily unbalanced. Therefore, using a ratio instead of a distance-based metric has the advantage of creating a relative measure. For example, in a subgroup with few negative samples, a small absolute number of false positives can still yield a high FPR. This effect is captured by a ratio-based metric but may be obscured by an absolute, distance-based measure. Sec. 2.3 of the manuscript has been revised to include a better reasoning behind the used metrics.
> 2. We clarified in the manuscript: "\[K was\] determined via grid search from $\{5,7,9,10\}$ based on a joint analysis of classification accuracy disparities and resulting subgroup sizes." Generally, increasing K increases subgroup granularity but reduces the number of samples per subgroup. Beyond a certain K, additional subgroups become empty or carry a negligible number of samples. We, therefore, selected K at the saturation point where performance differences stabilize while subgroup sizes remain sufficiently large for reliable analysis.
> 3. We appreciate the reviewer's idea for methodological improvement and, thus, re-ran subgroup discovery with the proposed approach. When DOMINO is run on positive samples only, all subgroups show a bias percentage above 50% and, therefore, the bias and non-bias subgroups cannot be distinguished anymore. For negative samples only, the TPR cannot be calculated, because the number of TP is 0. Other metrics, such as the accuracy or Delta FPR do not clearly distinguish *No Bias & No Disease* from *Bias & No Disease* for bias level 0.7 (see: table 2 in the next comment).

---

> > ### Author Response · Authors · 2026-01-23
> >
> > **Table 1:** Comparison of original and mitigated classifier for bias level 0.5.
> >
> > | Model     | Acc. B | Acc. NB | Recall B | Recall NB | Prec. B | Prec. NB |
> > |-----------|--------|---------|----------|-----------|---------|----------|
> > | Original  | 0.87   | 0.87    | 0.88     | 0.88      | 0.87    | 0.87     |
> > | Mitigated | 0.87   | 0.88    | 0.91     | 0.90      | 0.84    | 0.85     |
> >
> >
> > **Table 2:** Excerpt from subgroup discovery for bias level 0.7 of two examples with high and low bias percentage. The input dataset to DOMINO contains negative disease cases only.
> >
> > | Subgroup       | # pos/neg samples | Percentage Bias | Accuracy | Delta FPR |
> > |----------------|-------------------|------------------|----------|-----------|
> > | Bias Group     | 0 / 52            | 0.60             | 0.92     | -0.05     |
> > | No Bias Group  | 0 / 21            | 0.09             | 0.90     | -0.03     |

---

### Author Rebuttal · Authors · 2026-01-23

**Rebuttal:**

Dear Reviewers,

We thank you for your time and valuable feedback and hope the revisions are satisfactory. Please find the revised manuscript including tracked changes attached.

**Supporting Material:**

/attachment/8a443e2cef872872559f6681a2cca95e7c9ac32d.pdf

---

### Meta-Review · Area_Chair_Ebyt · 2026-02-07

**Recommendation:** Accept (Poster)
**Confidence:** 3

**Metareview:**

Although the results are preliminary, all reviewers noted that the method is novel and that the problem investigated in the paper is compelling. The reviewers agreed that the core idea is sufficiently original and substantive to merit presentation at MIDL.

---

### Decision · Program_Chairs · 2026-02-13

Accept (Poster)